# Molecular and Morphological Divergence of Australian Wild Rice

**DOI:** 10.3390/plants9020224

**Published:** 2020-02-10

**Authors:** Dinh Thi Lam, Katsuyuki Ichitani, Robert J. Henry, Ryuji Ishikawa

**Affiliations:** 1United Graduate School of Agricultural Sciences, Iwate University, Morioka, Iwate 020-8550, Japan; lamiasvn@gmail.com; 2Institute of Agricultural Science for Southern Vietnam, District 1, Ho Chi Minh City 121, Vietnam; 3Faculty of Agriculture, Kagoshima University, 1-21-24 Korimoto, Kagoshima, Kagoshima 890-0065, Japan; ichitani@agri.kagoshima-u.ac.jp; 4Queensland Alliance for Agriculture and Food Innovation, University of Queensland, Brisbane QLD 4072, Australia; robert.henry@uq.edu.au; 5Faculty of Agriculture and Life Science, Hirosaki University, 3 Bunkyo-cho, Hirosaki, Aomori 036-8561, Japan

**Keywords:** *Oryza*, speciation, divergence, life history, phylogenetic relation, Australian continent

## Abstract

Two types of perennial wild rice, Australian *Oryza rufipogon* and a new taxon Jpn2 have been observed in Australia in addition to the annual species *Oryza meridionalis*. Jpn2 is distinct owing to its larger spikelet size but shares *O. meridionalis*-like morphological features including a high density of bristle cells on the awn surface. All the morphological traits resemble *O. meridionalis* except for the larger spikelet size. Because Jpn2 has distinct cytoplasmic genomes, including the chloroplast (cp), cp insertion/deletion/simple sequence repeats were designed to establish marker systems to distinguish wild rice in Australia in different natural populations. It was shown that the new taxon is distinct from Asian *O. rufipogon* but instead resembles *O. meridionalis.* In addition, higher diversity was detected in north-eastern Australia. Reproductive barriers among species and Jpn2 tested by cross-hybridization suggested a unique biological relationship of Jpn2 with other species. Insertions of retrotransposable elements in the Jpn2 genome were extracted from raw reads generated using next-generation sequencing. Jpn2 tended to share insertions with other *O. meridionalis* accessions and with Australian *O. rufipogon* accessions in particular cases, but not Asian *O. rufipogon* except for two insertions. One insertion was restricted to Jpn2 in Australia and shared with some *O. rufipogon* in Thailand.

## 1. Introduction

The *Oryza* genus is comprised of 23 species with varying genome compositions and ploidy levels [1]. The two cultivated species, *Oryza sativa* and *Oryza glaberrima,* belong to AA genome species, and their progenitors were wild *Oryza rufipogon* and *Oryza barthii*, respectively. The AA genome species are dispersed across the major continents and were once classified as a single species, *Oryza perennis*, comprising Asian, American, African, and Oceanian forms [2]. The Asian species, *O. rufipogon* represents different life histories and varies from annual to perennial. Their life history is a continuum with annual, intermediate, and perennial forms [3,4]. The American species, *Oryza glumaepatula* also varies from annual to perennial. African species, *O. barthii* and *Oryza longistaminata*, however, are exclusively annual and perennial types, respectively.

Oceanian species had been known as *O. perennis* (later changed to the current species nomenclature) including annual and perennial types as a continuum within a single species [3]. After rearrangement of the species classification, an annual type was defined as an Oceanian endemic species, *Oryza meridionalis* and the perennial form as *O. rufipogon* [4]. Their distributions in Australia are well studied [5,6]. Speciation of these species has been confirmed using retrotransposon insertions [7,8] and crossing ability [9,10,11].

In general, annual and perennial species have different adaptive strategies to allocate their energy resources [3,12]. Annual species tend to have higher seed productivity than perennial species. *O. meridionalis*, the Australian annual species, produces plenty of seeds and disperses these seeds. *O. meridionalis* inhabits ponds or the periphery of ponds, ditches, or lakes during the rainy season. Water levels in wild rice habitats recede and water in the peripheral areas of annual species disappears during the dry season [13]. Annual species produce large amounts of seed for the next generation. In contrast, the life history of Australian perennial species is similar to Asian perennial species except for a unique taxon known as Jpn2 or taxon B [6,14]. In addition, Jpn2 type wild rice exhibits different morphological and genetic characteristics [14]. Including the new wild rice type, Australian perennial and annual rice chloroplast (cp) genomes have been completely sequenced in order to understand the uniqueness in evolutionary relationships among other wild rice [15,16]. This showed that the cp genome of Australian *O. rufipogon*, Jpn1 (taxon A) has a closer relationship to *O. meridionalis* than to Asian *O. rufipogon*, although its nuclear type tended to show higher similarity to Asian *O. rufipogon.* Another perennial species, Jpn2 (taxon B), also shared similarity not only with the cp genome to *O. meridionalis* but also the nuclear type [14,17,18]. This analysis showed that all Australian wild rice shared some cp genetic similarity with *O. meridionalis*. Nuclear genomes in Australia showed huge variation never seen in Asian wild rice. These findings with ecological observations confirmed that there were two types of perennial rice. Their distribution in northern Queensland and their unique morphological traits were also reported [6,14].

In this paper, we further characterized these two taxa at morphological and reproductive levels, which enabled us to determine how they have diverged at the species level. Cytoplasmic markers to distinguish them were developed and variation among natural populations was evaluated. These findings will help to distinguish these taxa in field research for further analysis and also give clues to their evolutionary origins. Retro-transposable elements were also used to screen the species examined in this study. Some of these provide clear evidence of phylogenetic relationships because of the unique mechanism of transposition insertion.

## 2. Results

### 2.1. Morphological Features

Two types of Australian perennial wild rice were collected (Table 1). Based on our previous report [14], identifying two types of perennials: Jpn1 (taxon A) and Jpn2 (taxon B), morphological traits were able to be discriminated between the Australian perennials. Bristle cells have a thorn-like architecture along the awns (Figure 1). SEM enabled us to compare the density of these cells. They varied from 2.33 to 5.33 per 200 μm square among Asian wild rice (Table 2). In *O. meridionalis*, W1299 and W1300 had 12.67 and 14.67 per 200 μm^2^, respectively. The density in Jpn1 was similar to that in Asian wild rice. That in Jpn2 was similar to *O. meridionalis*. There were significant differences between the two groups, W1299/W1300/Jpn2 and W106/W0120/W0137/Jpn1. Other traits, such as anther length, suggested that Jpn2 shared short anthers with other annual accessions such as W0106 in *O. rufipogon*, and W1299 and W1300 in *O. meridionalis*.

### 2.2. Maternal Lineages

In order to trace maternal lineages, next-generation sequencing data obtained from Jpn1 and Jpn2 were used for re-sequencing and comparison with the Nipponbare complete cp genome sequence. More than 53 million reads were obtained from the two accessions. Two genome sequences of *O. meridionalis*, and *O. rufipogon* were added for comparison. In all cases, 100% coverage was achieved with 733 to 2002 mean depth. When the nuclear genome was used as a reference genome, 66%–88% coverage with 7.6 to 11.4 mean depth was obtained.

Simple sequence repeats were found at 20 loci in the cp genomes. Simple insertions or deletions (INDELs) were also found at 21 loci (Table 3). Two loci were not amplified, and six loci were not confirmed because of difficulty of primer design for these fragments. One region ranging from nucleotide 17,336 to 17,392 of the Nipponbare cp genome was amplified as a single amplicon because of its short size. In total, 29 insertions/deletions (INDELs)/simple sequence repeats (SSRs) in the cp genome were polymorphic. Australian rice accessions including *O. meridionalis*, Jpn1, and Jpn2 shared the same genotype at 26 out of the 29 loci developed by plastid INDELs and SSRs.

Five chloroplast markers, INDEL1, INDEL11, INDEL13, INDEL18, and INDEL19, represented polymorphisms among natural populations (Appendix A (Appendix A)). Plastid types were defined as distinct combinations of each genotype. In total, nine plastid types (Type 1 to 9, r1, and r2) with r1 and r2 types in the control *O. rufipogon*, were recognized. Asian *O. rufipogon* and *O. sativa* accessions, were obviously different from the Australian wild rices.

Three accessions in PNG *O. rufipogon*, W1235, W1238, and W1239, and W2109 in Australian *O. rufipogon* shared the Type 5 plastid type with *O. meridionalis*. W1230 in Papua New Guinea *O. rufipogon* shared the r2 plastid type with the Asian type. W1236 carried a unique plastid type. Jpn2 shared Type 1 with *O. meridionalis*. Other *O. meridionalis* in the core collection divided into three types, Types 1, 5, and 8. Only two types, Types 1 and 2, were detected in the Northern Territory and in Western Australia. Newly collected accessions from Queensland carried seven types. Five of them were newly detected.

### 2.3. Reproductive Isolation

Biological species can be detected by the pollen fertility of hybrids. Jpn1 and Jpn2 were crossed with Asian wild rice and *O. meridionalis*. Each F_1_ plant was grown in a greenhouse, and leaf samples were used to check whether they were hybrids originating from the cross. Anthers were taken to check pollen fertility by staining with I_2_–KI. Well-stained pollen grains were counted.

Seed fertility was also assessed but this may not reflect reproductive ability of the respective plants (Table 4). W0106, W0120, and W1299 showed more than 95% pollen fertility. However, except for W0120, they showed lower seed fertility of 19.5% in W0106 and 22.3% in W1299. The panicles were bagged to prevent out-crossing and this might explain the low seed fertility. In combinations with Jpn1 and Asian *O. rufipogon*, F_1_ plants with W0106 and W0120 had more than 90% pollen fertility. However, seed fertility was relatively low, similar to self-pollination of W0106 and W1299. We relied on data from pollen fertility rather than seed fertility and concluded that by this criterion, Jpn1 is related to Asian *O. rufipogon*, and that Jpn2 is not close to either *O. rufipogon* or *O. meridionalis*.

### 2.4. Unique Insertion of Retrotransposable Element in Jpn2

In total, six presumed insertions were confirmed only in the Jpn2 genome but not in Nipponbare (Table 5). Two *pSINE1* insertions were shared among Jpn2 and 19 *O. meridionalis* accessions. Another insertion amplified with Chr3-10559212-r (w/L) and pSINE1-L showed an insertion shared among Jpn1, Jpn2, and 19 *O. meridionalis* accessions (Figure 2). Chr1-4067055-f (w/L) and pSINE1-L amplified the same amplicons not only from Jpn2 and 19 *O. meridionalis* accessions but also with W0106, which originated in India, suggesting that some parts of the Jpn2 genome share the insertion with wild rice from India. No *O. rufipogon* accessions in the core collection except for W2266 and W2267 were tested because of lack of DNA, and 19 *O. meridionalis* showed these insertions. Results suggested that the insertion was probably shared among *O. meridionalis* and W0106. Chr3-10203820-f (w/L) can amplify with pSINE1-L only in Jpn2 and no other *O. meridionalis* showed any amplicons. In screening for the insertion among 30 *O. rufipogon* accessions in the core collection, W0180 and W1921, both of which originated from Thailand, showed amplicons. The insertion sequence in Jpn2 was screened from the raw reads and 53 bp were recovered. When aligned with *pSINE1*, 92.4% high similarity was retained. When *pSINE3* insertions were examined, three of the presumed insertions were amplified only among Jpn2 and 19 *O. meridionalis* accessions.

## 3. Discussion

### 3.1. Unique Morphological Traits in Australian Wild Rice

*O. rufipogon* is composed of a continuum of annual and perennial strains in Asia. They represent different life history traits related to the r-K strategy to maximize fitness [4,12]. Perennial and annual types are regarded as K- and r-strategists, respectively. Intermediates represented the r–K continuum. K selection works for individuals to increase their life span and r selection works to produce more offspring. Thus, perennials spend more energy on vegetative organs before the flowering stage. Annuals spend energy to produce more panicles and seeds. Because anther size is related to preference for outcrossing, perennials tend to carry longer anthers than annuals and produce more pollen to maximize the chance of outcrossing [3,4,12]. Such resource allocation was also confirmed in three Asian *O. rufipogon,* the Australian perennial Jpn1, and the Oceanian annual *O. meridionalis*. In order to adapt to the dry season, *O. meridionalis* plants produce many seeds and die after scattering their seeds. *O. meridionalis* has short anthers and slender panicles. The appearance of Jpn1 is similar to Asian *O. rufipogon*, with similar long anthers and open panicles. Our measurements also suggested a trend. In our previous paper, we reported that Jpn2 represents a perennial life history [14]. It generated shoots and roots from its stems to follow water in peripheral areas, growing to the inner side because of the shrinking water mass during the dry season. The morphological appearance of Jpn2 was quite different to Jpn1. Anthers length is a unique characteristic in the morphology of this type, being shorter in *O. meridionalis* [14]. *O. rufipogon* W0106, an annual type, also shared this short anther characteristic. In this study, we also demonstrated another morphological trait characteristic of Jpn2. Jpn2 has a high density of bristle cells along the awn. Genome sequencing also suggested that Jpn2 shared higher similarity to *O. meridionalis* than *O. rufipogon* [16]. These characteristics of Jpn2 infer that this species/taxon has diverged from *O. meridionalis*.

### 3.2. Maternal Variation

Cytoplasmic marker systems can be developed for the mitochondrial genome as suggested in this report. Other markers were also developed based on whole cp genome sequences. Whole cp genome sequences have been determined for several Australian accessions [15,16,17,18,19]. The maternal genome data clearly showed Jpn1 and Jpn2 shared high similarity to *O. meridionalis* with some variation. INDELs and SSRs were designed based on the cp genome sequences. Core collections and natural populations were examined to determine the distribution of cp variation. Higher variation was found among accessions in Queensland compared with others accessions collected from the Northern Territory and Western Australia. Variations in cp genomes were distinguished at high resolution using single nucleotide polymorphisms [17,18]. This study showed that easily scored INDELs and SSRs also detected higher diversity in the northern Queensland accessions. These marker systems with whole cp genome screening will provide more clues about the maternal relationships among these related species/taxa and how they diverged.

### 3.3. Reproductive Barriers Among Australian Wild Rice

Reproductive barriers among *Oryza species* including the Australian species have been confirmed and numerical characteristics supported speciation reproductive barriers [2]. *O. meridionalis* already has high genetic reproductive barriers and sterility detected in F_1_ lines of crosses with Asian wild rice. Even among *O. meridionalis*, some sterile lines were reported [9]. In our study, F_1_ between Jpn2 and *O. meridionalis* displayed reproductive sterility. Jpn2 in particular, developed a reproductive barrier with both Asian wild rice and *O. meridionalis*. Because the extent of the reproductive barrier corresponded to that of two different organisms, it is concluded that Jpn2 does not belong to *O. rufipogon* or *O. meridionalis*. We have not yet determined when they diverged from each other. Clade analysis of cp genomes suggested that a clade including *O. meridionalis* diverged at a date estimated as 0.86–11.99 million years ago [18]. Similar estimation has also been reported based on sequences among *Oryza* genus [19,20,21,22]. Such a long time since divergence has allowed the accumulation of quite diverse genomes in the north-eastern part of Australia and created Jpn2 and various wild rice found at the P5 site.

F_1_ hybrids between Jpn2 and *O. meridionalis* showed relatively high pollen fertility with *O. meridionalis*, although the F_1_ showed complete seed sterility. It was suggested that the divergence between these two plants is a relatively recent event compared with the divergence from other species.

### 3.4. Retrotransposable Elements

Retrotransposable elements are well known markers for examining evolutionary pathways. This is mainly due to the unique mechanisms of transpositions. Two retrotransposable elements, *pSINE1* and *pSINE3*, were recognized in species divergence among AA genome and between Asian wild rice and *O. meridionalis* [3,6,23]. These have offered researchers a powerful tool for phylogenetic analysis. On the other hand, when there are no genome sequences, new markers to distinguish particular genomes are not available. In fact, there was no genomic data on insertions in a novel taxon such as Jpn2. Thus, we established a screening methodology to extract retrotransposable elements from raw reads. Recent developments in sequencing technology offer huge numbers of reads to increase target sites. Even with our limited volume of data, we succeeded in picking up insertions in the Jpn2 genome. The uniqueness of Jpn2 was also found with an insertion of the *pSINE1* retrotransposon, which was detected in Jpn2 only and in none of the other accessions of *O. meridionalis*. Two accessions of *O. rufipogon* in Thailand may provide key information on how Australian wild rice originated. This tool will open the way to draw a more precise evolutionary pathway and to understand valuable genetic resources among wild rice.

## 4. Materials and Methods

### 4.1. Plant Materials

Wild rice was collected in Australia with permission from the Queensland government, EcoAccess. We developed these collections as de novo resources, which can be accessed repeatedly from the same site with accurate GPS data allowing us to reconfirm their life cycles. Successive observations were made from 2009 until 2011, and the life history traits at the collection sites were reconfirmed year by year. This field research was supported by overseas scientific research funds (JSPS) and collaborative research with the Queensland Herbarium and Queensland Alliance for Agriculture and Food Innovation (QAAFI), University of Queensland. Thirty populations were collected from their natural habitat. Observation of the ecological habitats and life cycle of each population in April 2008, August 2009, and September 2009 were used to determine their life history such as annual or perennial behavior especially for Jpn1 and Jpn2 populations. Jpn1 and Jpn2 were typical perennial sites and individuals survived as living plants in a swamp (Jpn1) or a pond (Jpn2). In order to compare these accessions with cultivated rice, *Oryza sativa*, and wild species, *O. rufipogon* (W0106, W0120, and W0137) and *O. meridionalis* (W1297, W1299 and W1300) were compared. All plant materials were grown in greenhouse conditions at Kagoshima University. Samples collected from nature were compared with Jpn1 and Jpn2 grown from seeds collected to compare environmental effects on anther length and lemma size. Jpn1 and Jpn2 were crossed with W0106, W0120, W1297, and W1299 to test these relationships. The density of bristle cells on the surface of awns per 100 μm^2^ was counted using a scanning electric microscope (JSM-7000F, JEOL co., Japan). A core collection derived from the National Bio-Resource (NBR) Project in Japan was kindly provided, as shown in Appendix A (Appendix A) [24].

### 4.2. Crossing and Fertility Test

Jpn1 and Jpn2 were crossed with Asian wild rice, W106 and W120, and to Australian *O. meridionalis*, W1299 and W1300. F_1_ plants were grown in a greenhouse at Kagoshima University. Anthers were taken and stained with I_2_–KI solution and well-stained grains were counted as fertile pollen. The remaining panicles were wrapped in paraffin bags to prevent outcrossing. Fully filled grains were counted to calculate seed fertility.

### 4.3. Data Mining from Whole Genome Sequences

Whole genome sequences of Jpn1 and Jpn2 were obtained using Illumina GAIIx to develop INDEL markers of the cpDNA and retrotransposon INDEL markers. The total numbers of pair-end reads were 52,087,744 for Jpn1 and 54,749,858 for Jpn2. Total nucleotides sequenced were 3.9 Gb for Jpn1 and 4.1 Gb for Jpn2. Mean depth was 2022 in Jpn1 and 2002.5 in Jpn2. With our draft data, we aligned these raw reads to the cpDNA of Nipponbare (GenBank: GU592207.1) using CLC-work bench genomics version 6.0. Several INDELs were grouped together to screen for using single PCR reactions with the designed markers listed in Table 5.

Retrotransposable elements *pSINE1* and *pSINE3* have been reported to be uniquely found in either species, *O. rufipogon* or *O. meridionalis* [7,8,23]. Consensus sequences of the 5’ and 3’ termini were used to design a consensus probe to screen the draft sequence data. Based on the alignment of *pSINE1* elements, consensus sequences were presumed [8]. Two probe sequences were applied for *pSINE1* to screen the data: CCA.CA.CTTGTGGAGCTAGCCGG, in which the periods indicate degenerate nucleotides, for the 5’ termini, and TAGGT.TTCCCTAATATTCGCG for the 3’ termini. These degenerate probes were applied to screen raw reads of Jpn1 and Jpn2. 5’ termini which were confirmed to carry AAGACCCCTGGGCATTTCTC as the complementary sequence. Then, internal sequences of the 5’ probe were obtained from the read to confirm whether it shares homology, ranging from 74% to 82%. In these cases, we adopted the outside of 5’ termini as flanking sequences of *pSINE1* insertions. 3’ termini carried TAG followed by a poly T stretch. We adopted the downstream sites as flanking sequences of *pSINE1* insertions.

Based on *pSINE3* family elements, *r3004*, *r3005*, *r3012*, and *r3024*, the 5’ terminal consensus probe GCCGGGAAGACCCCGGGCC was used to screen internal sequences. The internal sequences were used to design an internal probe, CTAGCTCAGCTTGTGCTA. In order to examine the flanking sequences of insertions, the consensus probe was applied. After confirming the 5’ end shared the 5’ terminus of *pSINE3*, the outside sequences from TTTCTC were regarded as pSINE3 insertions.

When multiple reads were obtained as single locations, we specified the genome position based on the Nipponbare genome and detected 14 and 16 insertions that did not overlap. Of these, 21 could be aligned to the Nipponbare genome without *pSINE1* insertions at the site. Flanking sequences in the Nipponbare genome were applied to design primers to amplify the presumed insertions of either *pSINE1* or *pSINE3*. Outward primers inside *pSINE1* or *pSINE3* were also designed as shown in Table 5. Preliminary screening was performed with Nipponbare, three *O. rufipogon*, W0106, W0120, and W0137, two *O. meridionalis*, W1299 and W1300, and Jpn1, and Jpn2.

### 4.4. Data Analysis

Dendrograms were constructed using the neighbor-joining method based on Nei’s unbiased genetic distances by Populations1.2.30 beta2 program, which was downloaded from http://bioinformatics.org/~tryphon/populations/#ancre_bibliographie. All dendrograms were drawn by the TreeExplorer software used to show and edit population dendrograms as supplied with MEGA [25].

## 5. Conclusions

These data suggested that Jpn2 (taxon B) may be a distinct new species belonging to the *Oryza* genus and isolated from other species by reproductive barriers.

## Figures and Tables

**Figure 1 plants-09-00224-f001:**
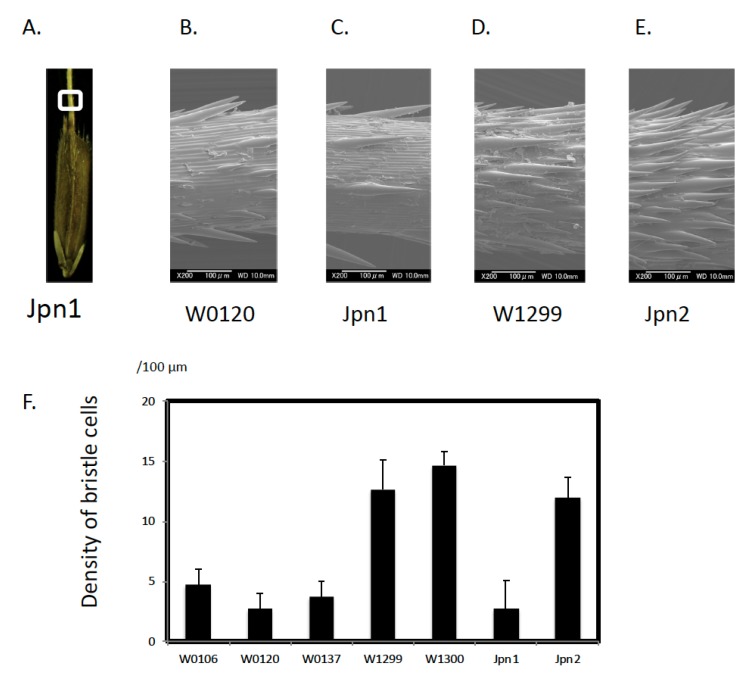
Variation in the density of bristle cells in awns. Panel **A**: spikelet of Jpn1, Panels B to E: enlarged SEM photos of W0120 (Panel **B**), Jpn1 (Panel **C**), W1299 (Panel **D**), Jpn2 (Panel **E**). Panel **F**: density of bristle cells per 200 μm^2^. Bars indicating standard error (n = 3).

**Figure 2 plants-09-00224-f002:**
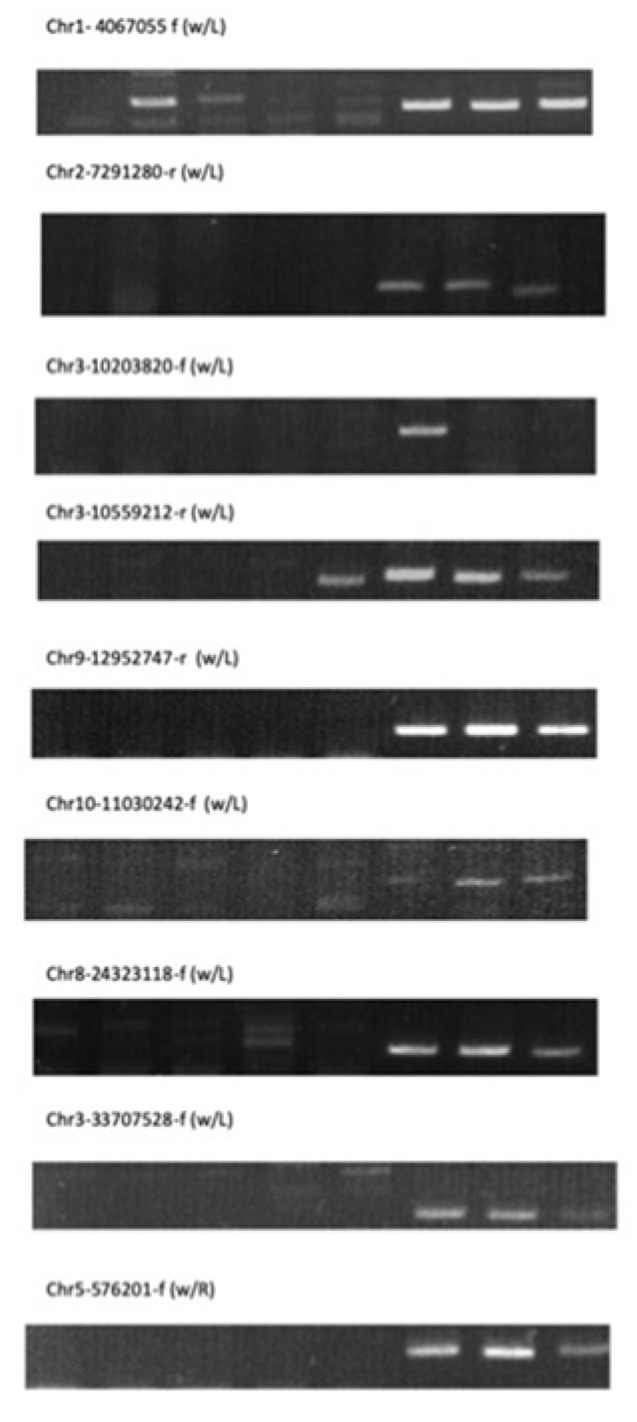
*pSINE1* and *pSINE3* insertions amplified with flanking primers and outward primers from *pSINE* consensus sequences. From lane 1 to 8, Nipponbare, W0106, W0120, W0137, Jpn1, Jpn2, W1299, and W1300 were used as each DNA template.

**Table 1 plants-09-00224-t001:** Samples collected in Australia and control core collections developed in NBRP.

		No. of	GPS Data		
Sites	Populations	Plants	S	E	Year
P26	P26a	8	S16.5332	E145.2138	2011
	P26b	8	S16.5529	E145.2136	2011
	P26c	8	S16.5541	E145.2130	2011
	P26d	8	S16.5539	E145.2128	2011
	P26e	8	S16.5536	E145.2125	2011
	P26f	8	S16.5536	E145.2124	2011
	P26g	8	S16.5535	E145.2123	2011
	P26h	8	S16.5534	E145.2122	2011
	P26i	8	S16.5533	E145.2122	2011
Jpn1	Jpn1	8	S16.3809	E145.1936	2011
Jpn2	Jpn2	8	S15.2622	E144.1239	2011
Jpn3	Jpn3	8	S15.0431	E143.4321	2011
P6	P6	8	S15 41519	E145 02473	2011
P7	P7E	8	S15 42003	E145 04219	2011
	P7L	8	S15 42003	E145 04219	2011
P8	P8W	4	S15 41302	E145 07237	2011
	P8R	4	S15 41302	E145 07237	2011
P10	P10H	8	S15.41416	E145.09040	2011
	P10L	8	S15.41416	E145.09040	2011
P12	P12	8	S15.31486	E144.22564	2011
P17	P17	8	S15.09256	E143.49481	2011
P21	P21	8	S15.23047	E144.07228	2011
P22	P22	8	S15.78099	E144.14333	2011
P23	P23	8	S15.31198	E144.22079	2011
P26j	P26j	8	S16.55118	E145.2136	2011
P26PL	P26PL	8	S16.15579	E145.2060	2011
P27	P27				
P5	P5O	8	S15 45329	E144 59419	2010
	P5N	8	S15 45329	E144 59419	2010
	P5W	8	S15 45329	E144 59419	2010
Core collection of NBRP					
*Oryza meridinoalis*		19*			
*Oryza rufipogon*		32			

*W1299 noted as no rank in Oryza database, was added to the core collection in this study.

**Table 2 plants-09-00224-t002:** Comparison of morphological traits, anther length, pancle length, paniclewidth, and density of bristle cells.

				Anther Length (mm)				Spikelet Length (mm)				Spikelet Width (mm)				Density of Bristle Cells (per 200 μm^2^)			
Species	Accession	Life History	n=	mean	±	SD		mean	±	SD		mean	±	SD		mean	±	SD	
*O.rufipogon*																			
	W0106	Annual	5	1.50	±	0.07	**	7.01	±	0.63	**	2.42	±	0.20		4.67	±	1.92	**
	W0120	Perennial	5	1.88	±	0.10	**	7.06	±	0.16	**	2.34	±	0.11		2.67	±	1.92	**
	W0137	Perennial	5	2.31	±	0.05	**	6.95	±	0.25	**	2.60	±	0.11		3.67	±	1.92	**
*O.meridionalis*																			
	W1299	Annual	5	1.42	±	0.05	**	6.47	±	0.27	**	2.32	±	0.12		12.67	±	3.42	
	W1300	Annual	5	ND				6.91	±	0.19	**	2.13	±	0.05	**	14.67	±	1.60	
Australian wild rice																			
Greenhouse																			
	Jpn1	Perennial	5	3.48	±	0.20		7.13	±	0.15	**	2.02	±	0.16	**	2.67	±	3.42	**
	Jpn2	Perennial	5	1.64	±	0.07	**	8.28	±	0.14	*	2.62	±	0.11		12.00	±	2.34	
Field																			
	Jpn1	Perennial	5	3.79	±	0.20		7.62	±	0.17	**	2.00	±	0.07	**	ND			
	Jpn2	Perennial	5	1.74	±	0.19	**	8.60	±	0.23		2.40	±	0.23		ND			

*,**: Significant differences compared with the longest anther of Jpn1, the largest panicle of Jpn2, the widest width of W0137, and density of bristle cells of W1300 at 5 and 1% levels, respectively.

**Table 3 plants-09-00224-t003:** INDEL and SSR markers in chloroplast genomes and developed markers.

	Genotype (Based on Relative Migration Distance)									INDEL Start Position in Nipponbare cp Genome		
Marker Type	Nipponbare	W0106	W0120	W0137	W1299	W1300	Jpn1	Jpn2	INDEL		Forward	Reverse
INDEL1	2	2	2	2	2	2	1	2	-C	1605	CTATTCCGAAGAGGAAGTCTAC	TCTCCGTATCAATGATCTGGTG
SSR1	1	1	1	1	2	2	3	2	+AA	3535	CTTTTGACTTTGGGATACAGTC	GATTAGTGCCTGATGTAGGG
INDEL2	2	2	2	2	1	1	1	1	-CAATC	5852	GGAATTTCCATCCTCAACAGA	GTTTTGTTACGGAAAAATGGTATG
SSR2	1	1	1	1	2	2	2	2	+A	6098	TTCTCGTATTTCTTCGACTCG	GATAAGAACTGCTCGTTAGATAG
INDEL3	1	1	1	1	2	2	2	2	+AGAAA	8192	GCCGCTTTAGTCCACTCAGCCATC	TCAATGCCTTTTTTCAATGGTCTC
SSR3	1	1	1	1	2	2	2	2	+A	11441	CTGGCTCGGTTATTCTATC	GAAAACCGGTATAGTTCTAGG
INDEL4	2	1	2	1	1	1	1	1	-AGGG	12669	GCAACAGGGTTCCCTAAACCG	GCCAAATTGAGCAGGTTGCG
INDEL5	2	2	2	2	1	1	1	1	-T	13566	GCTTCGCGACTCTGTACTCA	TACTTAAGGCGTCCTTAAGG
INDEL6	2	1	2	1	1	1	1	1	-AC	14011	GAAATCTGGGCCATAGAGAA	CTAAGCAGAGACATTCAGAATC
INDEL7	NA								-TATTTCTAAGA	14527	-	-
SSR4	2	2	2	2	1	1	1	1	-A	17099	GAAAAAATCCATGGAGGGAGAG	CCCAACATATCGCACATTTTCC
SSR5	2	1	1	1	3	3	3	3	-TTTCTA	17336	GGTCGCTTCTAGTAGCGATTATG	TGCCGAACTTTATTCTTTCTCTC
SSR6	including SSR5 to INDEL10								-TTTCTA	17358		
INDEL8									-ATAGAA	17379		
INDEL9									+AGAATTAT	17385		
INDEL10									+GAATTATATAGAAC	17392		
INDEL11	1	1	1	1	2	2	1	2	+TGG	19001	GAATATCATAAACTGTAAGTGGCAG	CACATGAAATTCTCGGGAACTCC
SSR7	1	1	1	1	2	2	2	2	+T	41464	GAGGCAAGTGTTCGGATCTATTATG	CTATATTATGCTCAAGGAAAGTAGA
SSR8	1	1	2	1	1	1	1	1	-TATAT	46086	CTCTAATTCGCAAATCTATTTTTC	CAAGAAATTCGCATGTTCTC
SSR9	2	1	2	1	1	1	1	1	-T	46174	GAGAACATGCGAATTTCTTG	CATACTATAACGCTTGATATTC
INDEL12	1	2	1	2	2	2	2	2	+T	47211	GTCGTGAGGGTTCAAGT	CGAGTTAATAATCGACATTCCTTGCC
INDEL13	1	1	1	1	2	2	1	2	+AGGAC	50351	GCCTGTCCAGTCTATAAACAAG	GGGTCTTTGAAACAGTTCG
SSR10	1	1	1	1	2	2	2	2	+T	53999	CATAGAATGTACACAGGGTGTACCC	CTCACAACGACAGGGTCTAC
INDEL14	INDEL14 and SSR11								-CTTTTTTTTTAGAATA	57017	GGATAGAAAGGCCGCGAG including	GACTATTGTATTTTTGAGTTTGC
SSR11									-A	57061		
INDEL15	2	2	2	2	1	1	1	1	-CTTTTCAAT	64815	CCAGATGCTTTGTCATTCCC	TCATGACTCTAAGGTCCAACC
INDEL16	1	1	1	1	2	2	2	2	+TTCCTATTTAATA	65452	GTCGTTATTGTCGTAAGCATACGA	GATGAATACCCTCGATACATATG
SSR12	NA								+T	65615	-	-
INDEL17	1	1	1	1	2	2	2	2	+t	66896	CCAATGGCTTTTGCTACTATAACC	GAAAGAAAGGGCTCCGGTG
SSR13	1	1	1	1	2	2	2	2	+A	71377	GCACCTGTTATCTCTATCAAG	GTCTGGTTGCGAGGTCTGAATAG
SSR14	2	2	2	2	1	1	1	1	-TTTCTA	75980	GATATCCGTTTCAGGGTAAA	CTGATTCGTAGGCGTGGAC
SSR15	2	1	2	1	1	1	1	1	-A	76232	CAAATTTTACGAACAGAAGCTC	CCGAAGACTCGAAGGATACC
SSR16	NA								-T	76574	CATAACTAAACCCTCGAAAGTAA	CCCGCCTATAGCGGTAATC
INDEL18	2	3	2	3	1	1	1	1	-T	77728	GCTACATTTAAAAGGGTCTGAGG	CTGCCAGCAAAATGCCC
SSR17	NA								-T	78423	-	-
INDEL19	1	1	1	1	2	2	2	2	+T	80090	GGGTTGTACCAAGTCTGAA	GCTCGAGGACGTAGTTCTCCCATAA
SSR18	NA								-C	93004	-	-
SSR19	1	1	1	1	2	2	2	2	+C	93534	GTTCGTCCTCAATGGGAAAATG	GGGAAGTCCTATTGATTGCTG
INDEL20	1	2	2	2	2	2	2	2	+AACA	104530	GATCATTTTCTGGCGTCAGCG	GAATATTGTACCGAGGAATTCG
INDEL21	1	1	1	1	2	2	2	2	+G	121618	AAGGCTCGAATGGTACGATC	CTTCTCGAGAATCCATACATCCC
SSR20	NA								-G	122142	-	-

NA: not amplified.

**Table 4 plants-09-00224-t004:** Pollen and seed fertility of self pollinated plants and F_1_ plants among Asian *O. rufipogon*, *O. meridionalis* and alternative perennials in Australia.

Pollen and Seed Fertilitty (%)									
Female	Male	Self Pollinated		Crossed with Jpn1		Crossed with Jpn2		Crossed with W1297	
		Pollen	Seed	Pollen	Seed	Pollen	Seed	Pollen	Seed
W0106		97.2	30.1	87.2	15.7	10.2	0.0	0.1	0.0
		98.8	8.9	95.7	18.3	0.8	0.0	0.1	0.0
	Mean*	98.0	19.5	91.4	17.0	5.5	0.0	0.1	0.0
W0120		98.1	96.8	84.0	24.5	10.2	0.0	21.3	0.0
		98.7	74.5	78.3	21.6	12.7	0.0	20.2	0.0
	Mean	98.4	85.6	81.2	23.1	11.5	0.0	20.8	0.0
W1299		95.7	20.0	0.0	0.0	29.4	0.0	99.3	42.9
		96.1	24.5	4.5	0.0	39.8	0.3	-	-
	Mean	95.9	22.3	2.2	0.0	34.6	0.1	-	-
W1297		95.7	54.4	5.3	0.0	53.2	0.0	-	-
		96.9	75.9	6.4	0.0	33.8	0.0	-	-
	Mean	96.3	65.2	2.2	0.0	34.6	0.1	-	-

*Mean: data obtained from multiple plants were averaged and noted the mean.

**Table 5 plants-09-00224-t005:** Screening retrotransposable element insertions.

Flanking Primers to Confirm			Insertion (+) / no-Insertion (-)	
*pSINE1*/*3* Insertions		Genome	
with Inner Primers toward Outside	Sequence	Position	Nipponbare	W0106	W0120	W0137	Jpn1	Jpn2	W1299	W1300	Remarks
*pSINE1*											
Jpn2-meridionalis class											
Chr2-7291280-r (w/L)	TCTCTCTACAGATAATGCTC	7291535	-	-	-	-	-	+	+	+	Other *O. meridionalis*
Chr9-12952747-r (w/L)	CACACCCATCTACATCGATG	12953007	-	-	-	-	-	+	+	+	Other *O. meridionalis*
Chr10-11030242-f (w/L)	GATTGCCGGCTTCTTTACTAG	11029860									
Australia class											
Chr3-10559212-r (w/L)	ACCTATAACAACTGAGAGAC	10559538	-	-	-	-	+	+	+	+	Other *O. meridionalis*
Jpn2-meridionalis-W0106 class											
Chr1-4067055-f (w/ L)	GAAAGAGATCACAGGTAAAC	4066713	-	+	-	-	-	+	+	+	Other *O. meridionalis*
Jpn2-W0180,W1921 class											
Chr3-10203820-f (w/L)	TCCACCGACTTATAAATCAC	10203450	-	-	-	-	-	+	-	-	No other *O. meridionalis*, but two *O. rufipogon*,W0180, W1921
Inner primer toward outside											
pSINE1-L	GAAGACCCCTGGGCGTTTCT										
(paired primer to amplify *pSINE1* insertion)											
*pSINE3*											
Chr3-33707528-f (w/L)	GTGTAAATATGTATTGTACC	33702014	-	-	-	-	-	+	+	+	Other *O. meridionalis*
Chr8-24323118-f (w/L)	GCCTATTACTATCAATCACC	24322813	-	-	-	-	-	+	+	+	Other *O. meridionalis*
Chr5-576201-f (w/R)	GATAACTAGGGTAAATGAC	575868		-	-	-	-	+	+	+	Other *O. meridionalis*
Inner primer toward outside											
pSINE3-R	TCCTTCCTAGATTGGTCCC										
pSINE3-L	TGCTAGCCGGGAAGACC										
(paired primer to amplify *pSINE3* insertion)

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
