# Peer review of "Molecular and Morphological Divergence of Australian Wild Rice"

_plants, 2020, doi:10.3390/plants9020224_

Round 1

Reviewer 1 Report

I have gone through the manuscript titled "Molecular and morphological divergence of Australian wild rice" and made some suggestions.

Intro: You may need to add more related citations in the introduction. 

Please describe the aim of the study clearly at the end of the introduction section. 

R&D: The results section is a bit long so that it is difficult to follow and understand. That would be good if the authors could make it shorter and focus on the main findings. In contrast, discussion section is too short and needs to be expanded. 

M&M: Growth conditions should be described in detail. 

"4.2. Morphological" trait is just a sentence, you may want to merge it with "4.1"

Ref: Reference list needs to be revised, there are different references with different format. 

Author Response

Dear Reviewer 1;

1st of all, thank you or sharing time to review our paper. I would like to revise based on all reviewers' comments.

Intro: You may need to add more related citations in the introduction. 

Reply; yes I tried to add additional citations.

Please describe the aim of the study clearly at the end of the introduction section. 

Reply; yes. Our point is 

characterize further the new taxon and apply developed method to know evolutionary pathway o the probable new species.

We added "In this paper, we further characterized the two taxa at morphological level and at reproductive level, which results enable us to know how they are diverged as species level. Cytoplasmic markers to distinguish them were developed and variation among natural variation was evaluated. It will help to distinguish them in field researches for further analysis and also give clues how they shared evolutionary pathways. Retro-transposable elements were also added to screen among species examined in this study. Components of the insertions will offer clear phylogenetic relationships because of the unique mechanism of transpositions."

R&D: The results section is a bit long so that it is difficult to follow and understand. That would be good if the authors could make it shorter and focus on the main findings. In contrast, discussion section is too short and needs to be expanded. 

Reply: we deleted redundant parts such as

1) mitochondrial sequence divergence and phylogenetic trees.

2)nuclear markers to identify the new taxa with other natural populations.

And discuss other parts to more for readers to understand the core part of our result.

M&M: Growth conditions should be described in detail. 

Reply: We added.

"4.2. Morphological" trait is just a sentence, you may want to merge it with "4.1"

Reply: Yes we revised.

Ref: Reference list needs to be revised, there are different references with different format. 

Reply: we added some references and changed the format to adapt to this journal.

Reviewer 2 Report

The introduction section needs to be revised to better state the objectives and significance of this study.

On line 57, “r type” needs to be defined here, as this is the first time it appears in the main text of manuscript.

Table 1, the GPS data for Jpn6-1.8k and Jpn6-3.1k were reported as “km distant from Jpn2-0k". However, Jpn2-0k was not on the list. Could this actually refer to Jpn6-0k? To better visualize the distribution of the populations studied, please consider presenting the locations on a map with the surrounding geographic features all marked.

In Table 2 caption, “levels” was misspelled as “leverls”.

Table 3 is missing caption. The accessions “W106”, “W120” and “W137” were named “W0106”, “W0120” and “W0137” here. Please keep them consistent. What is the distinction between “NA” and “ND”? If the primers failed to amplify the region, then it is not worthy putting their sequence in the table. What do the numbers in column 2 to 9 mean?

Line 109 to 121, were these results referring to Table S4 in the supplementary data? If so, please refer to it at appropriate positions in the text.

Line 124, if the 100 SSR loci used for phylogenetic analysis were the ones listed in Table S3, please refer to it in the main text.

The image of Figure 4 appears low in resolution.

Author Response

1st of all thank you for sharing time to review our paper. We tried to revise our paper with other comments from other reviewers also.

Major revised parts are

we deleted mitochondrial genome data, nuclear genotyping data. Because, cp genome markers are major points and nuclear divergence was described by SNPs data in previous papers. We decided not to mention further for other markers. Thus we deleted related figures and tables. Some of them are given comments to revise. Sorry to rearrange our results. I showed the revised version as attached iles.

The introduction section needs to be revised to better state the objectives and significance of this study.

Reply:we tried to revise.

On line 57, “r type” needs to be defined here, as this is the first time it appears in the main text of manuscript.

Reply: we simplify note the new taxon as Jpn2 and also referred another way to refer the type in other papers published by my colleague in this manuscript inside.

Table 1, the GPS data for Jpn6-1.8k and Jpn6-3.1k were reported as “km distant from Jpn2-0k". However, Jpn2-0k was not on the list. Could this actually refer to Jpn6-0k? To better visualize the distribution of the populations studied, please consider presenting the locations on a map with the surrounding geographic features all marked.

Reply: actually, the site was started from Jpn6 previously published and recorded the GPS. However, we simplified our data, and deleted redundant populations as shown in new table 1.

In Table 2 caption, “levels” was misspelled as “leverls”.

Reply; we revised.

Table 3 is missing caption. The accessions “W106”, “W120” and “W137” were named “W0106”, “W0120” and “W0137” here. Please keep them consistent. What is the distinction between “NA” and “ND”? If the primers failed to amplify the region, then it is not worthy putting their sequence in the table. What do the numbers in column 2 to 9 mean?

Reply; we revised. NA and ND were tried to integrate into one term NA not ampliied to further research.

Numbers are "Genotype (based on relative migration distance)". we added the explanation at the top.

Line 109 to 121, were these results referring to Table S4 in the supplementary data? If so, please refer to it at appropriate positions in the text.

We changed some parts of result but as for the result of genotyped plastid composition we referred Table S2 (plastid type).

Line 124, if the 100 SSR loci used for phylogenetic analysis were the ones listed in Table S3, please refer to it in the main text.

Reply: to simplify the data, we deleted nuclear data by using 100 loci. 

The image of Figure 4 appears low in resolution.

Reply: We deleted the result.

Reviewer 3 Report

I have read the Lam et al. manuscript 3 times, and I still do not really grasp the significance of their work.  I understand that they are looking at uncharacterized wild rice collections from Australia, and that one of them (Jpn2) appears to be either a novel species or a species that is diverging from O. meridionalis via a reproductive barrier. But the conclusions about the origin of the plastid vs. nuclear genomes need to be stated more clearly. I appreciate the effort expended on this study, but the phylogenetic analyses need to be improved.  In Figure 2, was the tree calculated from mitochondrial SNPs, or was it from cp data as described in the text? And 'Niipponbare' should be labeled as O. sativa spp. japonica. Figure 3 clearly shows two main clades, but the tree is unrooted, and it doesn't contain enough accessions.  And the accessions included are not labeled as to their species affinities.  

A major criticism is that, in this day and age, high-throughput DNA sequencing is available at a minimal cost. The phylogenetics in this manuscript would be much more meaningful if nuclear SNPs were used.  This would allow a large group of polymorphic loci to be developed that could then be used to partition the wild rice accessions with much higher resolution than what we see in Figs. 2, 3, and 4.  A PCA based on genetic distance would cluster the accessions in a meaningful way, showing true species and/or population affinities.  But as I said above, more plant accessions need to be included. You could sequence representative collections from each location rather than include every wild collection.  And because the genome is so small, you can combine many samples into a single sequencing lane and identify the reads from each individual library by their different barcodes.  10-20X depth should be enough for this kind of study since you have the 'Nipponbare' genome sequence to use as a template.

Why did you use the Populations 1.2.30 for your analyses, rather than one of the recent releases of MEGA? I have never heard of Populations 1.2.30, but MEGA is very well known and has been cited thousands of times.

You say that you calculated observed and expected heterozygosity, but I do not see these numbers in the manuscript.

When looking at plastid lineages, haplotype networks can be very informative.  You can analyze your cpDNA data in the manner.

The quality of the English in this manuscript is a major barrier to reading and understanding the research described within.  There is a huge number of words that are used improperly or misspelled, grammatical errors, and confusing sentences. After the phylogenetic analyses are repeated and the figures re-drawn, the authors MUST engage the services of a professional scientific manuscript editor to make sure that it can be read by a wide audience of plant biologists.

Author Response

1st of all thank you for sharing time to review our paper. We tried to revise our paper with other comments from other reviewers also.

Major revised parts are

we deleted mitochondrial genome data, nuclear genotyping data. Because, cp genome markers are major points and nuclear divergence was described by SNPs data in previous papers. We decided not to mention further for other markers. Thus we deleted related figures and tables. Some of them are given comments to revise. Sorry to rearrange our results. I showed the revised version as attached iles.

 Based on the comments, I would like to ask English editing service before uploading manuscript.

Before that, I would liketo explain the purpose.

Even the higher amounts of raw data could be available, sometimes limitation of budget is limited. Actually our data was extracted from the previous raw data of NGS at that time. However, we tried to develop extraction method to detect retrotransposable elements restricted into the new taxon and find the unique insertions by using core collections.

In wild rice, core collecton is highly reliable because Dr Morishima followed by Dr Kurata established with enormous data to select typical representatives. We also added natural populations. Out newly developed markers are highly effective with limited efforts and also even limited budgets. Thus it may support other researchers to select valuable materials to do further research. I hope these new tools will be noticed to other reseachers with some new findings. 

Population is not minor one. MEGA also can work. However, in this manuscript, nuclear data was deleted because of other reviewers (they want to simplify the data). Population can work with SSR genotypes but not sequence. Thus, the kinds of researchers prefer to use.

Authors

I have read the Lam et al. manuscript 3 times, and I still do not really grasp the significance of their work.  I understand that they are looking at uncharacterized wild rice collections from Australia, and that one of them (Jpn2) appears to be either a novel species or a species that is diverging from O. meridionalis via a reproductive barrier. But the conclusions about the origin of the plastid vs. nuclear genomes need to be stated more clearly. I appreciate the effort expended on this study, but the phylogenetic analyses need to be improved.  In Figure 2, was the tree calculated from mitochondrial SNPs, or was it from cp data as described in the text? And 'Niipponbare' should be labeled as O. sativa spp. japonica. Figure 3 clearly shows two main clades, but the tree is unrooted, and it doesn't contain enough accessions.  And the accessions included are not labeled as to their species affinities.  

A major criticism is that, in this day and age, high-throughput DNA sequencing is available at a minimal cost. The phylogenetics in this manuscript would be much more meaningful if nuclear SNPs were used.  This would allow a large group of polymorphic loci to be developed that could then be used to partition the wild rice accessions with much higher resolution than what we see in Figs. 2, 3, and 4.  A PCA based on genetic distance would cluster the accessions in a meaningful way, showing true species and/or population affinities.  But as I said above, more plant accessions need to be included. You could sequence representative collections from each location rather than include every wild collection.  And because the genome is so small, you can combine many samples into a single sequencing lane and identify the reads from each individual library by their different barcodes.  10-20X depth should be enough for this kind of study since you have the 'Nipponbare' genome sequence to use as a template.

Why did you use the Populations 1.2.30 for your analyses, rather than one of the recent releases of MEGA? I have never heard of Populations 1.2.30, but MEGA is very well known and has been cited thousands of times.

You say that you calculated observed and expected heterozygosity, but I do not see these numbers in the manuscript.

When looking at plastid lineages, haplotype networks can be very informative.  You can analyze your cpDNA data in the manner.

The quality of the English in this manuscript is a major barrier to reading and understanding the research described within.  There is a huge number of words that are used improperly or misspelled, grammatical errors, and confusing sentences. After the phylogenetic analyses are repeated and the figures re-drawn, the authors MUST engage the services of a professional scientific manuscript editor to make sure that it can be read by a wide audience of plant biologists.

Reviewer 4 Report

Present manuscript Molecular and morphological divergence of Australian wild rice seems to be a straightforward effort aiming to characterize wild rice through molecular markers. But the data presented is too confusing, hard to understand for non-expert readers because authors used multiple terms/ short forms at different places, for example, Abstract- line 16-17 Oryza rufipogon (r) type (Taxon A) and a O. meridionalis (m) type (Taxon B).... again they extend it in line 62-63 :-Taxon A (Jpn1 type) and Taxon B (Jpn2 type).

-The manuscript needs to present a simple and understandable manner for common readers. Use proper as well as uniform terms.

The introduction must be extended it is not covering all aspects of a present experiment like molecular markers, current status wild rice divergence/ characterization, etc.,   Exactly what is the number of SSRs used? find 95, 33, 100

As such, the key question boils down to if the current work presents enough new information to warrant publication in this journal. It might, but the way the information is packaged and presented probably needs to be changed to help accentuate the important points

Author Response

1st of all thank you for sharing time to review our paper. We tried to revise our paper with other comments from other reviewers also.

Major revised parts are

we deleted mitochondrial genome data, nuclear genotyping data. Because, cp genome markers are major points and nuclear divergence was described by SNPs data in previous papers. We decided not to mention further for other markers. Thus we deleted related figures and tables. Some of them are given comments to revise. Sorry to rearrange our results. I showed the revised version as attached iles.

Present manuscript Molecular and morphological divergence of Australian wild rice seems to be a straightforward effort aiming to characterize wild rice through molecular markers. But the data presented is too confusing, hard to understand for non-expert readers because authors used multiple terms/ short forms at different places, for example, Abstract- line 16-17 Oryza rufipogon (r) type (Taxon A) and a O. meridionalis (m) type (Taxon B).... again they extend it in line 62-63 :-Taxon A (Jpn1 type) and Taxon B (Jpn2 type).

Reply: we simpified our terms. Jpn2 we adopted for this purpose. Because our colleagues use different terms in previous papers, we referred the terms inside manuscript. However, we mostly use Jpn2 in this report.

-The manuscript needs to present a simple and understandable manner for common readers. Use proper as well as uniform terms.

Reply: We tried and revise.

The introduction must be extended it is not covering all aspects of a present experiment like molecular markers, current status wild rice divergence/ characterization, etc.,   Exactly what is the number of SSRs used? find 95, 33, 100

Reply: we revised the parts. SSRs actually 100 loci at final stage. However, we deleted nuclear data to simplify our data, we did not show anymore.

As such, the key question boils down to if the current work presents enough new information to warrant publication in this journal. It might, but the way the information is packaged and presented probably needs to be changed to help accentuate the important points

Thanks

Reviewer 5 Report

This paper deals with diversity found in Australian wild rice taxa (Oryza spp.). The authors conducted many experiments and analyses such as morphological-, molecular phylogenetic-, and even crossing test to examine reproductive barrier among each taxon. My suggestion is mainly going on how to present their findings.

Figure 4. I could not understand the relationship between panel A and panel B. Why don’t they combine and present one NJ tree? To do so, Australian Oryza’s diversity is clearly shown.

The authors seem to use several words indicate a certain group or biological unit, I guess, but it is very confusing. Judging from the sentences, I think they are the same Taxon A= r type= Jpn1, Taxon B= m type=Jpn2, but am I correct or not?

Also, accession number presents three or four digits (e.g. W106, W120, W137 in L73, W0106 in L74). It is also confusing whether W106=W0106 means same accession or not.

L208 As anther size is related to preference for outcrossing, perennials tend to carry longer anthers than annuals…. Please cite references to support this idea.

L210 Such resource allocation (=bearing longer anthers?)was also confirmed in three Asian O.rufipogon, the Australian perennial, Jpn1, ----- with Jpn2.

There is a contradiction in this sentence. According to the result provided in table2, length of anther in Jpn2 is very short.

L212 --- die after shattering their seeds. ?? --die after scattering their seeds?

L229 Twelve types were recognized ---- geographical distribution was shown (Figure 2).

Figure 2 is phylogenetic tree using mtDNA. Wrong citation with Table S4? But even if it is true, there is no geographical distribution.

L238 Why and how Jpn2 developed a reproductive barrier so rapidly? Discuss here if the authors have some speculation.

Author Response

1st of all thank you for sharing time to review our paper. We tried to revise our paper with other comments from other reviewers also.

Major revised parts are

we deleted mitochondrial genome data, nuclear genotyping data. Because, cp genome markers are major points and nuclear divergence was described by SNPs data in previous papers. We decided not to mention further for other markers. Thus we deleted related figures and tables. Some of them are given comments to revise. Sorry to rearrange our results. I showed the revised version as attached iles.

Suggestions for Authors

This paper deals with diversity found in Australian wild rice taxa (Oryza spp.). The authors conducted many experiments and analyses such as morphological-, molecular phylogenetic-, and even crossing test to examine reproductive barrier among each taxon. My suggestion is mainly going on how to present their findings.

Figure 4. I could not understand the relationship between panel A and panel B. Why don’t they combine and present one NJ tree? To do so, Australian Oryza’s diversity is clearly shown.

Reply: We tried to integrate all accessions into one tree. However, it is too hard to see. Thus in previous paper, we 1st positioned Jpn1 and Jpn2, 2nd panel showed how the relations among natural populations with Jpn1 and Jpn2. 

However, to simplify our data, we omitted nuclear data and focus on plastid type and how to disting

The authors seem to use several words indicate a certain group or biological unit, I guess, but it is very confusing. Judging from the sentences, I think they are the same Taxon A= r type= Jpn1, Taxon B= m type=Jpn2, but am I correct or not?

Reply: We decided to integrate our term as Jpn1 or Jpn2 to refer the unique accession we collected at the site Jpn1 and Jpn2. And referred the relation with Taxon A and Taxon B in manuscript. Because our colleagues used such terms in different papers, we once tried but reconsidered now. 

Also, accession number presents three or four digits (e.g. W106, W120, W137 in L73, W0106 in L74). It is also confusing whether W106=W0106 means same accession or not.

Reply: yes. We revised.

L208 As anther size is related to preference for outcrossing, perennials tend to carry longer anthers than annuals…. Please cite references to support this idea.

Reply: We added citations. Oka and Morishima 1967 Evolution 21:249-258 et al in this manuscript. 

L210 Such resource allocation (=bearing longer anthers?)was also confirmed in three Asian O.rufipogon, the Australian perennial, Jpn1, ----- with Jpn2.

Reply: Jpn1 is probably corresponded to Morishima et al.'s Australian rufipogon. Because I have belonged to the laboratory and our core collection was provided fro m National Institute of Genetics, which were materials they used,

As for Jpn2, it showed discrepancy. Thus we noted the unique characteristics.

There is a contradiction in this sentence. According to the result provided in table2, length of anther in Jpn2 is very short.

Reply: Jpn2 has unique trait. Perennial but shared most of traits with annual. We referred the inconsistency in discussion.  We speculate Jpn2 diverged from O. meridionalis annual species. Thus they shared all genomes and also some traits except for larger seed size. We also detected bristle cell in one of the traits.

L212 --- die after shattering their seeds. ?? --die after scattering their seeds?

Reply: we revise.

L229 Twelve types were recognized ---- geographical distribution was shown (Figure 2).

Reply: we did not add the corresponded figure but only table. At this time, we described the geographical tendency in Table S2 (revised version, previous Table S4). 

Figure 2 is phylogenetic tree using mtDNA. Wrong citation with Table S4? But even if it is true, there is no geographical distribution.

Reply: Figure 2 was mitochondrial genome sequence based tree. But this time, we deleted the data.

L238 Why and how Jpn2 developed a reproductive barrier so rapidly? Discuss here if the authors have some speculation.

Reply: we added cp genome's divergence time estimated. it ranged

" long divergence date estimated ranging from 0.77 to 7.62 myr [20]."

Thus within this time span, such speciation could be available.

" Similar estimation has been also reported based on sequences among Oryza genus [20, 21, 22]. Such a quite long time allowed to accumulate quite diverse genomes in north-east part of Australia and created Jpn2 and various wild rice found in P5 site."

Round 2

Reviewer 3 Report

I appreciate that the authors followed the advice of myself and another reviewer and removed data to focus on the cpDNA sequences and the retrotransposon insertions.  However, the newly-added text is very difficult to understand. 

Author Response

I would like to thank you for reviewing our report.

We asked English editing service and also try to simplify our manuscript. I hope you will fell better. We also followed the comments of other reviewers.

We tried to focus on cp genome data.

Reviewer 4 Report

I still do not really understand the significance of the present work. The revised manuscript I just reduced version of the previous one without mitochondrial genome data, nuclear genotyping data. I think the accessions included are not enough not to make final conclusion. Again with revised manuscript also very confusing, hard to understand.

Author Response

I would like to thank you for reviewing our paper.

As for limited number of wild rice in nature, our materials are also restricted. However, the most significant accession is Jpn2. We tried to characterize the novel accession. In addition, this time we carefully check our manuscript and also ask ed English editing service as well.

Reviewer 5 Report

I admitted the author revised following my suggestions. However, I have to say there are still many points to be improved in their ms. 

L18.  Jpn2 type is distinct by its larger spikelet size but O.meridionalis like.....    Many of the reader don't know O.mridionalis is annual. Add "annual species" or "annual" before "O.meridionalis".

L28. One insertion is restricted to Jpne in ---      What it Jpne?  or It means Jpn2?

L54. Annual species produce large amounts of seed(s) for the next...

L56. except for unique taxon known as Jpn2 and taxon B...            as Jpn2 or taxon B?

L61. O.meridionalis than to "Asian??" O.rufipogon...   Only O.rufipogon, we could not understand what the authors mention.

L65. genomes showed huge variation(s?) never seen in Asian wild rice(s?).

L66. observation confirmed that there were two types of perennials (in Australia?).

L70. I could not understand the meaning of following sentense: "Cytoplasmic markers to distinguish them were developed and variation among natural variation was evaluated."

L85. W106/W120/W137/Jpn1   -----  W0106/W0120/W0137/Jpn1

Table 4  Wh species does W1297 belong to?

legend of Table 4. ---O. meridionalis and alternative perennials in  "??" in Australia?

L121 Only two types, Type 1 1 and 2, ----.  Only two types, Type 1 and 2??

L140 Unique insertion of Retrotransposable element sin Jpn2   ----        what is "sin"? or "in"??

L203 Reproductive barrier among Oryza species....         Oryza species

L207. I can't understand what they want to say in the following sentence: In our study, Jpn2 type identity compared with other species are still required to .....

L213. I could not find the figure 0.77 to 7.62 myr in Kim et al. 2015?? Rather, judging from the branch length, the divergent time of Australian Oryza is less than 0.3.

I wanted to know “how” and “why” Jpn 2 developed reproductive isolation mechanism against other Australian relatives, e.g., using different pollinators or take different pollination mechanism (wind vs. insects), not to ask how long have it took. If the authors have no idea, they don’t need to add this sentence. Anyhow, “the length of divergent time“ is not only the way to develop reproductive barrier.

L225. mainly due to uniue ----         due to unique....

L247. Thirty natural populations were collected.          Samples were collected from thirty natural populations.???

L310. taon B     ---   taxon B

Author Response

We would like to thank you for your kind comments. We revised the parts based on your comments. In addition, we took English editing service to improve our manuscript in detail. 

We reply to reviewer's comment as follow.

I admitted the author revised following my suggestions. However, I have to say there are still many points to be improved in their ms. 

L18.  Jpn2 type is distinct by its larger spikelet size but O.meridionalis like.....    Many of the reader don't know O.mridionalis is annual. Add "annual species" or "annual" before "O.meridionalis".

Reply: We added as “annual species, O. meridionals

L28. One insertion is restricted to Jpne in ---      What it Jpne?  or It means Jpn2?

Reply: it was misspelled. We revised.

L54. Annual species produce large amounts of seed(s) for the next...

L56. except for unique taxon known as Jpn2 and taxon B...            as Jpn2 or taxon B?

Reply: Annual species produce large amounts of seed for the next generation. In contrast, the life history of Australian perennial species is similar to Asian perennial species except for a unique taxon known as Jpn2 or taxon B

L61. O.meridionalis than to "Asian??" O.rufipogon...   Only O.rufipogon, we could not understand what the authors mention.

Reply: O. meridionalis than to Asian O. rufipogon, although its nuclear type tended to show higher similarity to Asian O. rufipogon.

L65. genomes showed huge variation(s?) never seen in Asian wild rice(s?).

Reply: Nuclear genomes in Australia showed huge variation never seen in Asian wild rice.

L66. observation confirmed that there were two types of perennials (in Australia?).

Reply: These findings with ecological observations confirmed that there were two types of perennial rice.

L70. I could not understand the meaning of following sentense: "Cytoplasmic markers to distinguish them were developed and variation among natural variation was evaluated."

Reply:

“Cytoplasmic markers to distinguish them were developed and variation among natural populations was evaluated.” In order to know natural variation, cytoplasmic markers were developed.

L85. W106/W120/W137/Jpn1   -----  W0106/W0120/W0137/Jpn1

Reply: We revised.

significant differences between the two groups, W1299/W1300/Jpn2 and W106/W0120/W0137/Jpn1.

Table 4  Wh species does W1297 belong to?

Reply: W1297 belongs to O. meridinoalis. Thus, F1 with O. rufipogon showed complete male sterility.

legend of Table 4. ---O. meridionalis and alternative perennials in  "??" in Australia?

Reply :

L121 Only two types, Type 1 1 and 2, ----.  Only two types, Type 1 and 2??

Reply: we revised as “Only two types, Type 1 and 2”

L140 Unique insertion of Retrotransposable element sin Jpn2   ----        what is "sin"? or "in"??

Reply: revised as “2.4. Unique insertion of retrotransposable element in Jpn2”

L203 Reproductive barrier among Oryza species....         Oryza species

L207. I can't understand what they want to say in the following sentence: In our study, Jpn2 type identity compared with other species are still required to .....

Reply: “In our study, F1 between Jpn2 and O. meridionalis represented reproductive sterility. Jpn2 particularly developed a reproductive barrier with both Asian wild rice and O. meridionalis.”

L213. I could not find the figure 0.77 to 7.62 myr in Kim et al. 2015?? Rather, judging from the branch length, the divergent time of Australian Oryza is less than 0.3.

Reply: Tang et al.??

Clade analysis of cp genomes suggested that a clade including O. meridionalis diverged at a date estimated as 0.77–7.62 million years ago [20].

I wanted to know “how” and “why” Jpn 2 developed reproductive isolation mechanism against other Australian relatives, e.g., using different pollinators or take different pollination mechanism (wind vs. insects), not to ask how long have it took. If the authors have no idea, they don’t need to add this sentence. Anyhow, “the length of divergent time“ is not only the way to develop reproductive barrier.

Reply: We demonstrated whether Jpn2 belongs to known species or novel one. In the process, diverse maternal lineages are found in northern Queensland. As climate condition is quite severe, small populations are isolated each other. Under the conditions, self pollination is quite adaptive. In rice, as self-compatibility has not developed. Therefore outcrossing is one of methods but not only the way to reproduction for perennial species. Outcrossing will happen by wind occasionally with some parts of self-pollination.

Geographical isolation will gradually developed speciation between Jpn2 and O. meridionalis. As Jpn2 shares quite same genetic traits with O. meridionalis, quite long duration will develop its speciation. To consider speciation, the time span is also important. Thus, we would like to keep some references to note how old O. meridionalis is,

L225. mainly due to uniue ----         due to unique....

Reply: We revised as “This is mainly due to the unique mechanisms of transpositions.

L247. Thirty natural populations were collected.          Samples were collected from thirty natural populations.???

Reply: We revised as “Thirty populations were collected from natural habitat.”

L310. taon B     ---   taxon B

 Reply: We revised

Round 3

Reviewer 4 Report

Make F1 uniform.it should be like F1 Table 4 - complete the title/legend Recheck/rephrase line 125-126 Newly... Line 111- what is the mean of genotypes here? Table 3 -denote the means of 1 and 2.  line 149- O. meridionalis 19 accessions; table 1--18 accessions??

Need fine redrafting for minor errors like above.

Author Response

I would like to appreciate to give kind comments.Based on the comments, we revised as following.

We here use only F1 but not F1.

L111 The genotype meant plastid INDELs and SSRs. So we described as

"Australian rice accessions including O. meridionalis, Jpn1, and Jpn2 shared the same genotype at 26 out of the 29 loci developed by plastid INDELs and SSRs."

F1 fertility in Table 4. We added footnote as "*Mean : data obtained from multiple plants were averaged and noted the mean.".

Table 1 and L149: we added one accession, W1299 to the core collection (18 accessions) supplied from Oryza base. We noted the accession in Table 1 and manuscript.